# Inducible Systemic *Gcn1* Deletion in Mice Leads to Transient Body Weight Loss upon Tamoxifen Treatment Associated with Decrease of Fat and Liver Glycogen Storage

**DOI:** 10.3390/ijms23063201

**Published:** 2022-03-16

**Authors:** Jun Liu, Shuya Kasai, Yota Tatara, Hiromi Yamazaki, Junsei Mimura, Seiya Mizuno, Fumihiro Sugiyama, Satoru Takahashi, Tsubasa Sato, Taku Ozaki, Kunikazu Tanji, Koichi Wakabayashi, Hayato Maeda, Hiroki Mizukami, Yasuhiro Shinkai, Yoshito Kumagai, Hirofumi Tomita, Ken Itoh

**Affiliations:** 1Department of Stress Response Science, Center for Advanced Medical Science, Hirosaki University Graduate School of Medicine, 5 Zaifu-cho, Hirosaki 036-8562, Japan; cmu_liuj@outlook.com (J.L.); ytatara@hirosaki-u.ac.jp (Y.T.); yamazaki.hiromi@fbri.org (H.Y.); jmimura@hirosaki-u.ac.jp (J.M.); tsubasa_s@outlook.jp (T.S.); itohk@hirosaki-u.ac.jp (K.I.); 2Laboratory Animal Resource Center, University of Tsukuba, 1-1-1 Tennodai, Tsukuba 305-8575, Japan; konezumi@md.tsukuba.ac.jp (S.M.); bunbun@md.tsukuba.ac.jp (F.S.); satoruta@md.tsukuba.ac.jp (S.T.); 3Laboratory of Cell Biochemistry, Department of Biological Science, Graduate School of Science and Engineering, Iwate University, 4-3-5 Ueda, Morioka 020-8551, Japan; tozaki@iwate-u.ac.jp; 4Department of Neuropathology, Institute of Brain Science, Hirosaki University Graduate School of Medicine, 5 Zaifu-cho, Hirosaki 036-8562, Japan; kunikazu@hirosaki-u.ac.jp (K.T.); koichi@hirosaki-u.ac.jp (K.W.); 5Faculty of Agriculture and Life Science, Hirosaki University, 3 Bunkyo-cho, Hirosaki 036-8561, Japan; hayatosp@hirosaki-u.ac.jp; 6Department of Pathology and Molecular Medicine, Hirosaki University Graduate School of Medicine, 5 Zaifu-cho, Hirosaki 036-8562, Japan; hirokim@hirosaki-u.ac.jp; 7Environmental Biology Laboratory, Faculty of Medicine, University of Tsukuba, 1-1-1 Tennodai, Tsukuba 305-8575, Japan; ya_shinkai@md.tsukuba.ac.jp (Y.S.); yk-em-tu@md.tsukuba.ac.jp (Y.K.); 8Department of Cardiology and Nephrology, Hirosaki University Graduate School of Medicine, 5 Zaifu-cho, Hirosaki 036-8562, Japan; tomitah@hirosaki-u.ac.jp

**Keywords:** GCN1, tamoxifen toxicity, body weight loss, VLDL, lipid storage

## Abstract

GCN1 is an evolutionarily-conserved ribosome-binding protein that mediates the amino acid starvation response as well as the ribotoxic stress response. We previously demonstrated that *Gcn1* mutant mice lacking the GCN2-binding domain suffer from growth retardation and postnatal lethality via GCN2-independent mechanisms, while *Gcn1*-null mice die early in embryonic development. In this study, we explored the role of GCN1 in adult mice by generating tamoxifen-inducible conditional knockout (CKO) mice. Unexpectedly, the *Gcn1* CKO mice showed body weight loss during tamoxifen treatment, which gradually recovered following its cessation. They also showed decreases in liver weight, hepatic glycogen and lipid contents, blood glucose and non-esterified fatty acids, and visceral white adipose tissue weight with no changes in food intake and viability. A decrease of serum VLDL suggested that hepatic lipid supply to the peripheral tissues was primarily impaired. Liver proteomic analysis revealed the downregulation of mitochondrial β-oxidation that accompanied increases of peroxisomal β-oxidation and aerobic glucose catabolism that maintain ATP levels. These findings show the involvement of GCN1 in hepatic lipid metabolism during tamoxifen treatment in adult mice.

## 1. Introduction

The ribosome-interacting protein GCN1 is conserved from yeast to human and is essential for the amino acid starvation response (ASR) [1]. Amino acid starvation induces the activation of the protein kinase GCN2 in a GCN1-dependent manner, which in turn phosphorylates the translation initiation factor eIF2α and attenuates global 5′ cap-dependent translation. The translation of a subset of transcripts including yeast Gcn4 and mammalian-activating transcription factor 4 (ATF4) is selectively enhanced by eIF2α phosphorylation via the suppression of inhibitory upstream open reading frames (uORFs). The increased Gcn4/ATF4 expression, in turn, stimulates the expression of genes that are involved in amino acid synthesis and transport, thereby counteracting amino acid starvation.

Previously, we found that *Gcn1* is essential not only for ASR but also for embryonic growth using two strains of *Gcn1* mutant mice. The mice with the deletion of *Gcn1* exons 46–53, which encode the domain that interacts with GCN2 via the RWD domain (RWDBD) (hereafter called *Gcn1*^Δ*RWDBD*^ mice), showed embryonic growth retardation and postnatal lethality [2]. The mice with deletion of the *Gcn1* exon 2 (hereafter called *Gcn1^null^* mice), which causes a premature stop codon at the twelfth residue, exhibited a severer phenotype and earlier embryonic lethality than the *Gcn1*^Δ*RWDBD*^ mice [2]. Intriguingly, these embryonic phenotypes were not observed in the *Gcn2*-KO mice [3,4], which indicates that GCN1 regulates a GCN2-independent pathway that is important for embryonic growth and development. The *Gcn1*^Δ*RWDBD*^ mouse embryonic fibroblasts (MEFs) showed decreased cell proliferation and G_2_/M arrest, whereas *Gcn2*-KO MEFs did not show this phenotype [2]. Interestingly, the deletion or downregulation of DRG2, a heterodimer partner of another RWD-containing gene, DFRP2, caused similar cell cycle arrest and embryonic growth retardation phenotypes [5,6]. Moreover, several recent studies have identified the GCN2-independent and GCN20-dependent functions of GCN1 in several species, including apoptosis regulation in *Caenorhabditis elegans* [7] and immune regulation in *Arabidopsis thaliana* [8]. In mammals, it has been reported that halofuginone, an inhibitor of glutamyl-prolyl tRNA synthetase, inhibits inflammation in a GCN1-dependent but GCN2-independent manner [9].

Wu et al. have also suggested that GCN1 may play an important role in conveying the stress kinase signals during translation elongation stress, probably via recognizing colliding ribosomes [10]. In fact, cryo-electron microscopic demonstration of yeast GCN1 that is directly bound to a disome (two ribosomes colliding) indicated the role of GCN1 during translation stress in recognizing ribosome collision [11]. In addition, yeast GCN1 suppresses frameshift at the ribosome stall site of the CGA codon repeat [12]. In terms of the regulation of translational elongation, GCN2 plays important roles in protecting the mice with rare codon-mediated neuronal degeneration [13] and in rare codon-mediated translational feedback in neurospora [14], although the role of GCN1 in these situations was not established. Thus, in addition to its established role in ASR, GCN1 may be important for relaying the more widespread stress signals that are initiated from the ribosome.

Since standard *Gcn1* knockout leads to embryonic lethality or premature death, the role of GCN1 in adult mice has not been examined. To clarify the functional role of GCN1 in adult mice, the present study used the tamoxifen-induced Cre/loxP system to generate systemic *Gcn1* exon 2 conditional KO (CKO) in adult mice. The *Gcn1* CKO mice showed transient body weight loss and decreased energy source in the liver, blood, and white adipose tissue without affecting the food intake or viability. An alteration in serum VLDL as well as liver proteome indicated that GCN1 regulates lipid metabolism in the liver under tamoxifen treatment.

## 2. Results

### 2.1. Establishment of Gcn1 CKO Mice

To generate tamoxifen-inducible *Gcn1* CKO mice, loxP sites were inserted into the *Gcn1* exon 2 flanking regions (Figure 1A). The floxed (*Gcn1^fl^*^/*fl*^) mice were crossed with *ROSA26*-targeted *CreERT2* knock-in (*R26^Cre^*^/*Cre*^) mice, which express a fusion protein of Cre recombinase and the ligand-binding domain of estrogen receptor that is activated by tamoxifen metabolites but not by endogenous ligands [15]. The deletion of *Gcn1* exon 2 results in a frame shift in exon 3 and premature termination after the 12th codon (Figure 1A, mouse GCN1 protein). The male homozygous *Gcn1* floxed *CreERT2* knock-in (*Gcn1^fl^*^/*fl*^::*R26^Cre^*^/*Cre*^) mice were administrated 10 intraperitoneal (i.p.) tamoxifen injections (75 mg/kg/day, two rounds of five consecutive days with a 2-day interval) (Appendix A). The floxed mice without the *CreERT2* (*Gcn1^fl^*^/*fl*^::*R26^+/+^*) were administrated with tamoxifen and used as negative controls (NC). Genotyping of *Gcn1* exon 2 by PCR was used to determine both the wild-type and floxed alleles (Figure 1B). The PCR primers targeting the outside of the two loxP sites detected the KO allele in CKO mice (Figure 1B).

Next, we analyzed the GCN1 protein expression in various tissues of wild-type male mice. GCN1 was ubiquitously expressed in wild-type mice, but was especially abundant in the liver, pancreas, intestine, and testes, and lowly expressed in the heart and skeletal muscle (Figure 2A). *Gcn1* CKO efficiency was compared among the untreated wild-type (WT), NC, and CKO mice that were dissected two days after the cessation of tamoxifen injections (day 14; Figure 2B–K). GCN1 expression was dramatically decreased in the liver, intestine, and colon, and to a lesser extent in the heart, kidney, pancreas, spleen, and testes. No alteration in the brain GCN1 expression was observed, which is consistent with previous reports that *CreERT2* in the *ROSA26* locus was barely expressed in the brain [16]. Of note, the liver GCN1 expression in the NC mice was lower than that in the wild-type mice. As this lower expression was also observed by injection of the control vehicle, it was due to the vehicle injection but not to the loxP insertion or tamoxifen toxicity (Appendix A). Since the expression of GCN1 in WT and NC before tamoxifen injection (day 0) was comparable in the liver, heart, and kidney (Appendix A), the insertion of loxP in the *Gcn1* exon 2 was unlikely to affect the basal GCN1 expression. We also confirmed the defective ASR response and proliferation in *Gcn1* KO MEFs using the using *Gcn1^fl^*^/*fl*^::*R26^Cre^*^/*Cre*^ MEFs and tamoxifen treatment (Appendix A).

### 2.2. Tamoxifen-Induced Body Weight Loss in Gcn1 CKO Mice

To investigate the effect of *Gcn1* deletion in CKO mice, we monitored the body weight, food and water intake, and urine and fecal excretion each day during the tamoxifen injections period, and every five days after the last injection for 30 days (until day 42). The tamoxifen administration resulted in a robust body weight loss in the CKO mice, which reached the maximum level at the end of the injections, and then partially recovered (Figure 3A). Despite the body weight reduction, the mice looked healthy and maintained normal activity even at day 14. Conversely, the NC mice’s body weight did not decrease, except for the day after the first injection. Compared with the NC mice, the CKO mice showed a comparable food intake and a significant increase in water intake, especially after the cessation of the tamoxifen injections (Appendix A). Although the urine and fecal excretion by the CKO mice showed an increased tendency, abnormal excretion such as diarrhea and bloody stools was not observed (Appendix A). In addition, the CKO mice survived for at least 10 months without any abnormality (data not shown). As it has been reported that Cre recombinase activation itself causes detrimental effects [17,18], we compared tamoxifen-induced body weight changes of *CreERT2* knock-in (*Gcn1^+^*^/*+*^::*R26^Cre^*^/*Cre*^) mice with those of CKO and NC mice (Appendix A). *CreERT2* knock-in mice showed similar body weight changes to those of the NC mice.

To analyze the tamoxifen-induced body composition changes, tissue weight changes in the *Gcn1* CKO and NC mice were examined at day 14 and day 42. We found that in contrast to the NC mice, in the CKO mice, the liver and epididymal WAT (eWAT) weight that was normalized to body weight was decreased at day 14 and recovered at day 42 (Figure 3B,F). Changes in the other tissues were comparable between the genotypes and tended to decrease with time (Figure 3C–E,G). As it is conceivable that the cells that escaped recombination, and thus express GCN1, proliferate faster, thereby replacing the *Gcn1* CKO cells, we examined the GCN1 expression levels in the liver at day 42. Although the GCN1 expression in the liver of the NC mice recovered almost to the level of day 0 by day 42, we did not detect any GCN1 expression in the CKO mouse liver at day 42 (Appendix A).

### 2.3. Combination of GCN1-Deletion and Tamoxifen-Treatment Induced the Rapid Loss of Body Weight Gain

To determine whether the body weight loss of tamoxifen-induced *Gcn1* CKO was caused by increased sensitivity against tamoxifen in the *Gcn1* CKO mice, the CKO and NC mice were subjected to a second round of tamoxifen injections at day 43 and the body weight change was monitored until day 84 (Figure 4A). The *Gcn1* CKO mice that were injected with tamoxifen showed a significant body weight loss that was similar to the body weight change during the first round of injections (Figure 4A). Conversely, body weight loss was not observed in the NC mice that were injected with either tamoxifen or vehicle as well as in the CKO mice that were injected with vehicle, which indicates that the *Gcn1* CKO mice were specifically susceptible to tamoxifen-induced body weight loss. The mice were dissected at day 56 (2 days after the second round’s last injection) and day 84 (30 days after the last injection) and the weight of several tissues was analyzed. Although a statistically significant difference between NC and CKO was not detected in the WAT weight at day 56, a transient decreasing tendency was observed (Figure 4B–D). Additionally, a significant decrease in the testes weight was observed in the CKO mice from day 42 (Figure 4E).

To determine whether the tamoxifen metabolism is different between the NC and CKO mice, the serum concentrations of tamoxifen and its metabolites, 4-hydroxytamoxifen (4-OHTA) and endoxifen, were analyzed by LC-MS/MS. All the metabolites were detected only at day 14, and the concentrations of the metabolites were not significantly different between the CKO and NC mice, except for (E)-endoxifen which was lower in CKO than NC (Appendix A).

### 2.4. Tamoxifen Administration Drastically Impaired Glucose and Fat Metabolism in Gcn1 CKO Mice

We tested mouse serum for various tissue injury markers (Appendix A). Although the difference was not statistically significant, an increasing tendency was observed in the liver toxicity markers, aspartate aminotransferase (AST) and alanine aminotransferase (ALT), at day 14 in the CKO mice compared with the NC mice. However, no obvious pathological features were observed in the CKO liver by H&E staining (Appendix A). Blood glucose was significantly decreased in the CKO mice at day 14 and recovered to a level that was comparable to that in the NC mice at day 42 (Figure 5A). The serum non-esterified fatty acid (NEFA) was also decreased significantly in the CKO mice at day 14 compared to the NC mice (Figure 6C). Furthermore, a rapid decrease in the glycogen content was observed in the CKO liver at day 14 by PAS staining (Appendix A) and a quantitative kit (Appendix A), although the decrease was not statistically significant. No difference in the glycogen content and distribution was observed at both day 14 and day 42. At day 0, the glycogen concentration in the NC livers (24.7 ± 4.1 µg/mg) tended to be lower than that in the livers of the CKO (50.8 ± 8.2 µg/mg), wild-type (66.6 ± 9.6 µg/mg), and CreERT2 knock-in mice (60.4 ± 8.3 µg/mg) for unknown reasons (Appendix A and data not shown).

It has been reported that tamoxifen treatment associates with liver steatosis in mice in a mouse strain-dependent manner [19,20,21] as well as with the risk for fatty liver disease in breast cancer patients with long-term treatment [22,23]. Oil Red O staining showed that lipid droplets were transiently decreased in the CKO mice at day 14 (Figure 5C,D). To determine whether systemic starvation is induced by *Gcn1* CKO and tamoxifen treatment, we analyzed serum glucagon and insulin levels and autophagy in the muscle; however, no significant alteration was detected (Appendix A). In addition, the blood ketone bodies were not detected (data not shown). A significant elevation in the gene expression of the gluconeogenic gene, phosphoenolpyruvate carboxykinase (*PEPCK*), was observed in the CKO liver at day 14 compared with day 0 (Appendix A). To determine whether *Gcn1* CKO can affect nutrient absorption in the intestine, the fecal glucose was quantified (Figure 5B). The fecal glucose was significantly decreased in the NC mice, but not in the CKO mice, at day 14 (Figure 5B) suggesting the probable increase of glucose uptake only in the NC mouse intestine.

To determine whether the intestinal lipid absorption and subsequent lipoprotein production are perturbed in the CKO mice, we determined the lipid content in the chylomicron and lipoprotein fractions. Profiling of the serum lipoprotein revealed that both the triglyceride and cholesterol in the chylomicron fraction were comparable between the NC and CKO mice (Figure 6A,B). Both cholesterol and triglyceride in VLDL were significantly decreased in the CKO mice at day 14 compared with the levels at day 0, suggesting that a decrease in VLDL production in the liver can result in decreases in serum NEFA (Figure 6C) and glucose (Figure 5A), as well as in white adipose tissue weight (Figure 3F). The HDL cholesterol was significantly decreased in both the NC and CKO at day 14, which can account for the change in the total cholesterol (Figure 6A), whereas the total triglyceride change roughly reflects the VLDL triglyceride content (Figure 6B). Despite histological analysis of the intestine, no pronounced abnormality was found (data not shown), supporting the assertion that the VLDL production in the liver can be primarily compromised, decreasing the systemic lipid supply and storage in the eWAT.

It has been reported that tamoxifen has adverse effects on mitochondrial functions, including the inhibition of mitochondrial respiration [17], mitochondrial DNA depletion [24], and uncoupling of the proton gradient [25,26]. However, the activity of COX and SDH, and the mitochondrial DNA content in the liver were not altered by tamoxifen treatment at day 14 (Appendix A). Additionally, the liver ATP contents were also maintained at day 14 (Appendix A). We also analyzed the oxygen consumption rate (OCR) using primary hepatocyte culture that was established from NC and CKO mice at day 42, which showed a tendency for OCR to increase in CKO hepatocytes with statistical significance only in the non-mitochondrial respiration (*p* = 0.016) (Appendix A). Although the hepatocytes were treated with 4-OHTA before OCR assay, clear data supporting the increase in proton leakage were not obtained (data not shown).

Consistent with the decrease in the eWAT weight, H&E staining showed that the adipocyte size was significantly decreased in eWAT but was comparable in the inguinal WAT (iWAT), and this difference was recovered at day 42 (Appendix A). Because tamoxifen has been reported to induce UCP1 expression in WAT [27], we examined the *Ucp1* expression in the *Gcn1* CKO WAT. To our surprise, *Ucp1* expression was significantly induced only in the iWAT of the NC mice, but not in the CKO mice at day 14 (Appendix A). *Ucp1* expression in eWAT and in brown adipose tissue (BAT) showed no significant alteration. We also analyzed the expression of genes that are associated with lipid metabolism such as adiponectin, lipoprotein lipase (LPL), hormone-sensitive lipase (HSL), carnitine palmitoyl transferase 1a (Cpt1a), acetyl-CoA carboxylase 1 (Acc1), PPARα, PPARγ, and SREBP1. No significant alteration was detected in the CKO and NC eWAT at day 14 (data not shown).

### 2.5. Liver Proteome Analysis Revealed Drastic Metabolic Remodeling Leading to the Inhibition of Mitochondrial FAO and Enhanced Peroxisomal FAO in Gcn1 CKO Mice

To obtain insight into the mechanism behind the *Gcn1* CKO mouse phenotype, we performed proteome analysis of the liver. LC-MS/MS analysis of liver protein identified 1349 proteins with a false discovery rate (FDR) < 1%, among which 60 proteins were significantly (*p* < 0.05) altered between the CKO and NC (Appendix A). A total of 137 proteins correlated with a discriminant model that was constructed in an orthogonal partial least square-discriminant analysis (OPLS-DA) with p_1_(corr) > 0.6 or < −0.6 were analyzed by IPA. This revealed the downregulation of estrogen biosynthesis (*p* = 5.45 × 10^−6^, z-score = −0.447) and the upregulation of oxidative phosphorylation (*p* = 4.05 × 10^−5^, z-score = 0.816) canonical pathways (Figure 7A). Then, provided that GCN1 is a translational regulator, among the 60 differentially expressed proteins we focused on the 19 whose protein levels were decreased by more than half, including Cpt1a, 3-methylcrotonyl-CoA carboxylase (MCC) subunits Mccc1 and Mccc2, and hydroxyacyl-CoA dehydrogenase trifunctional multienzyme complex subunit beta (Hadhb) (Figure 7B,D and Appendix A). The proteomic analysis appeared to be reliable as GCN1 ranked second among the 60 differentially expressed proteins. A decrease of elongation factor 1-beta (eEF1β) was also an interesting finding as we previously identified elongation factor 1-alpha as a GCN1-binding protein (Figure 7B and data not shown). Decreases of GCN1 and eEF1β may commonly contribute to the translation of these proteins as there were no changes in their mRNA levels (Figure 7C). This is consistent with a recent finding that GCN1 binds the collided ribosomes and may be essential for the translation of certain elongation-resistant proteins [11].

It is notable that Cpt1a and Hadhb are key proteins of long-chain fatty acid β-oxidation (FAO) in the mitochondria. Thus, it was expected that long-chain FAO in the mitochondria would be more affected in the tamoxifen-treated *Gcn1* CKO mice compared with its status in the tamoxifen-treated NC mice. Fatty acids are also metabolized in the peroxisomal FAO system [28]. In particular, fatty acid omega oxidation (ω-oxidation) in the peroxisome is enhanced when mitochondrial FAO is defective [29]. Consistent with these observations, there were increases in cytochrome P450 family (Cyp) 4 subfamily F member 3 (Cyp4f3) and Cyp4A14, which potentially mediate omega oxidation of fatty acids [30,31], and ATP binding cassette subfamily D member 3 (Abcd3), which is a transporter of both medium- and long-chain fatty acids [32] and dicarboxylic fatty acid-CoA [33] in the peroxisome. In addition, enoyl-CoA delta isomerase 2 (Eci2), which enhances mitochondrial peroxisomal contact, was increased [34], indicating enhanced crosstalk between mitochondrial and peroxisomal metabolism. Peroxisomal FAO metabolizes fatty acids to short-chain fatty acyl-CoA or acetyl CoA, which are further metabolized in the mitochondria to generate energy [29]. Consistent with this, there were increases in acyl-CoA dehydrogenase short/branched chain (Acadsb) and acyl-CoA dehydrogenase short chain (Acads), which metabolize short or branched fatty acyl-CoA as well as acetyl CoA in the mitochondria [35,36]. It is of note that similarly extensive metabolic remodeling involving peroxisomal activation was reported in the muscle of Cpt1bM knockout mice [37].

In addition to the changes in peroxisome enzymes, the fatty acid elongase enzyme (i.e., very long-chain enoyl-CoA reductase, Tecr), although it is not the rate-limiting enzyme in the fatty acid elongation cycle, was increased by 7.7-fold, indicating that the fatty acid metabolism in the ER was also increased. Mccc1, Mccc2, and Hmgcl are in the same pathway of leucine catabolism in the mitochondria [38,39]. Both Mccc1 and Mccc2 were significantly decreased while Hmgcl was increased. The induction of Hmgcl might be a reaction compensating for the decreases of Mccc1 and Mccc2. Deficiencies of both MCC subunits and Hmgcl cause inborn errors of metabolism in human [38,39]. Interestingly, deficiencies of MCC were reported to widely affect the transcriptome of mitochondrial energy metabolism, including those that are involved in electron transport and mitochondrial FAO [40]. Triglycerides that are used for VLDL generation in the liver are made by the lipolysis of lipid droplet triglycerides into free fatty acids and the re-esterification process [41], and carboxylesterase (Ces)3b and Ces2a play important roles in lipid droplet lipolysis [42,43]. As Ces3b was markedly increased by 13.2-fold in *Gcn1* CKO compared with the level in NC, it may explain the rapid decrease of lipid droplets in the *Gcn1* CKO. The liberated fatty acids may be used for peroxisomal fatty acid oxidation, but not for re-esterification and VLDL synthesis.

As speculated above, the fatty acids in the *Gcn1* CKO were likely metabolized by peroxisomal FAO. Thus, the energy of fatty acids appears to be dissipated by the peroxisomal FAO generating hydrogen peroxide instead of electron transport chain (ETC) substrates. However, despite this futile increase of peroxisomal FAO, ATP in the liver was maintained in the *Gcn1* CKO (Appendix A). This might have been achieved mainly by the increased flux of fatty acids into the peroxisome, resulting in an increase of short-chain acylcarnitine metabolism in the mitochondria. Furthermore, increased aerobic metabolism of glucose, represented by the increased protein expression of Slc25a1 and ETC components or related proteins, especially those of complex I, may help to maintain ATP in the liver. Peroxisomal FAO generates hydrogen peroxide (H_2_O_2_) using FADH_2_ as a reducing agent. Consistent with this, there was an increase in biliverdin reductase B (BLVRB), which is a flavin reductase for a variety of substrates [44].

## 3. Discussion

Tamoxifen-dependent CreERT2 is a useful tool for producing the conditional KO of floxed genes without Cre recombinase background activity and genotoxicity [16]. However, tamoxifen can exert both agonistic and antagonistic effects on endogenous estrogen receptors [45] and impair mitochondrial functions [26]. In the current study, there was clear tamoxifen-induced body weight loss in the *Gcn1* CKO mice, but not in the NC. A slight but not significant decrease in the body weight of CreERT2 knock-in mice (Appendix A) cannot explain the phenomenon that is manifested by *Gcn1* CKO. As Gcn1 expression in the brain did not differ between the CKO and NC mice (Figure 2C), the body weight loss is probably independent of the regulation by the central nervous system. The mitochondrial FAO and leucine catabolism in the liver appears to be primarily impaired by the combination of tamoxifen treatment and *Gcn1* CKO. Although the tamoxifen-induced liver steatosis and vacuolation have previously been reported [20,24,46], they were not observed in our study, probably due to the different age of the mice or the different experimental conditions.

In this study, for the first time we demonstrated that the combination of *Gcn1* CKO and tamoxifen treatment decreased the fat and glucose storage in the liver. Our unbiased proteomic analysis suggested that the downregulation of mitochondrial FAO leads to a set of compensatory responses to maintain energy production in the mitochondria. ETC defects lead to the downregulation of FAO via the inhibition of trifunctional enzyme as trifunctional enzyme is extremely sensitive to the decrease of NAD^+^/NADH upon mitochondrial ETC inhibition [47]. However, it was previously reported that the experimental dose of tamoxifen affects the mitochondrial function via a variety of mechanisms, including the uncoupling of OXPHOS and the decrease of membrane potential [26]. Tamoxifen inhibits ETC and FAO, among others. Indeed, Larosche et al. demonstrated that tamoxifen inhibits FAO and Cpt1a activity as well as hepatic triglyceride secretion in mice [24]. Also, tamoxifen inhibits cholesterol-5,6-epoxide hydrolase [48], Ces1d, and liver fatty acid binding protein (FABP) [49]. Since Ces1d mediates VLDL production [50], the disturbance of lipid metabolism might be caused by tamoxifen. Thus, we surmise that NC mice that were injected with tamoxifen exhibit inhibition of ETC and lipid metabolism, but the alteration of lipid metabolism might be significantly more exacerbated in *Gcn1* CKO. This should not be due to an increased concentration of tamoxifen metabolites in *Gcn1* CKO (Appendix A). GCN1 appears to have buffering or protective function against the tamoxifen-mediated mitochondrial dysfunction. Indeed, it was previously proposed that the nutrient-sensing network may play a role in the intricate communication between ETC and FAO defects [51]. Furthermore, it is of note that tamoxifen or even vehicle treatment decreased GCN1 protein levels in the liver (Appendix A). Thus, the decrease of GCN1 itself may contribute to the tamoxifen effect on liver mitochondrial function.

The rapid decrease of the liver lipids upon tamoxifen injection in *Gcn1* CKO in our study was impressive. Acetyl CoA and NADPH are important drivers of fatty acid synthesis. In line with the increase of peroxisomal ω-FAO Cyp450s, cytochrome P450 reductase that uses NADPH as a reducing agent was significantly increased in CKO (Appendix A). Furthermore, peroxisomal FAO generates hydrogen peroxides using FADH_2_ as reducing agents and there was an increase in BLVRB that reduces oxidized flavin using NADPH as a cofactor. Thus, it was expected that NADPH was scarce in *Gcn1* CKO upon tamoxifen treatment, leading to the inhibition of lipid synthesis. The pentose phosphate enzyme transketolase (Appendix A) was increased to compensate for the decrease of NADPH. Considering the marked increase of lipid droplet degradation enzymes Ces3b and Ces2a, both the inhibition of lipid synthesis and the enhanced degradation of lipids may contribute to the rapid disappearance of lipids from the liver.

The exact molecular mechanisms underlying the GCN1-deletion-mediated effects remain unclear. Although we identified the alteration in the liver proteome by *Gcn1* CKO and tamoxifen treatment, further study using liver-specific *Gcn1* CKO mice is needed to determine whether the proteome alteration in day 14 CKO liver and the mouse phenotype itself is a liver-intrinsic GCN1 effect or not. Furthermore, comparison of the day 14 and day 42 CKO liver proteome will distinguish the effect of GCN1 itself from the combination of *Gcn1* CKO and tamoxifen treatment on the liver proteome. GCN1 exerts its effects via both GCN2-dependent and -independent mechanisms [2,9]. However, it is notable that GCN2 deletion is protective against high-fat-induced accumulation of liver lipids and insulin resistance [52]. Furthermore, it was recently shown that a protective effect of GCN2 inhibition is dependent on Nrf2 and associated with the induction of peroxisomal FAO, indicating that GCN2 may act at least in part downstream of GCN1 [53]. Since leucine or valine enhances FAO in cardiomyocyte via GCN2 inactivation and derepression of ATF6-PPARα transcription cascade [54], this pathway may mediate the metabolic changes in the *Gcn1* CKO liver. The reason why the specific protein that is involved in mitochondrial FAO or leucine catabolism was decreased in *Gcn1* CKO (Figure 7B) is currently unclear. However, both fatty acids and branched chain amino acids including leucine are metabolized in the similar pathway and yields acetyl CoA or acyl-CoA [55]. Furthermore, leucine activates mTORC1 interconnecting leucine metabolism and central energy metabolism [56]. Thus, GCN1 may play an integral role in mitochondrial FAO or leucine metabolism. As the gene expression did not differ between NC and *Gcn1* CKO regarding Cpt1a, Hadhb, Eef1b, Mccc1, and Mccc2 (Figure 7C), the differences should occur at the post-transcriptional level. Actually, previous reports from studies on *C. elegans* demonstrated that GCN1 and ABCF3 are involved in the translational regulation of a wide range of genes [7]. Furthermore, the recent finding of GCN1 specifically binding to disome suggested that GCN1 may be involved in the translation of difficult codons [11]. Thus, GCN1 may regulate the translational elongation during tamoxifen treatment, although further analysis is required to clarify the molecular mechanisms that are involved in this. On the other hand, GCN1 was recently identified in Mccc2-interacting protein in human hepatocellular carcinoma [57]. Thus, GCN1 may regulate the translation or stability of Mccc2 and affect mitochondrial leucine metabolism, thus protecting against the toxicity of tamoxifen. GCN1-dependent multiple mechanisms may regulate leucine catabolism in the mitochondria.

The reason why the eWAT mass was specifically decreased upon tamoxifen treatment in *Gcn1* CKO is currently unclear. As GCN1 is significantly expressed in the eWAT, it may regulate adipose tissue differentiation or metabolism. A recent study demonstrated that DRG2, one of the GCN1 downstream candidates, is involved in adipose tissue differentiation in mice [58]. However, notably, mice with liver-specific *Cpt1a* knockout show a similar phenotype upon consumption of a high-fat diet (e.g., loss of body weight gain, loss of fat mass) [59]. It was demonstrated that FGF21 was involved in the liver-mediated decrease of adipose tissue. Therefore, it is speculated that some interorgan communication may occur, which might explain the eWAT decrease occurring in *Gcn1* CKO. Otherwise, decreases of blood VLDL and NEFA may affect the eWAT metabolism, but further studies are required to clarify this. Tamoxifen-induced *Ucp1* expression was unexpectedly observed in the NC iWAT, but not in the CKO mice (Appendix A). Thus, GCN1 may be specifically required in the lipid regulation in eWAT and transcription factors that are responsible for *Ucp1* expression, such as ATF2 and/or CREB [60], may be regulated by GCN1. Additionally, it is known that glucagon release is enhanced by hypoglycemia in pancreatic α-cells [61], but in this study the serum glucagon was not enhanced by hypoglycemia in *Gcn1* CKO at day 14 (Appendix A). As the histochemical analysis revealed that the number of pancreatic α-cells was not altered in the *Gcn1* CKO mice (data not shown), we speculate that GCN1 is involved in the response to hypoglycemia at other points of action. These probable GCN1-dependent effects should be studied in tissue-specific knockdown analysis in the future.

As GCN1 was most abundantly expressed in the testes (Figure 2A) and tamoxifen administration decreased the testis weight (Figure 4E), we analyzed the fertility of the *Gcn1* CKO male mice. Given that the mating of CKO male mice, after the recovery from tamoxifen injection, with CreERT2 knock-in female mice produced *Gcn1* heterozygous offspring roughly at a Mendelian ratio (data not shown), *Gcn1* may be dispensable for spermatogenesis. As even a low dose of tamoxifen adversely affects the reproductive system [45], the toxicity of tamoxifen toward the reproductive system of the *Gcn1* CKO mice should be carefully analyzed and compared with that in the NC and tamoxifen-injected CreERT2 knock-in mice.

In conclusion, tamoxifen-induced *Gcn1* CKO mice showed body weight loss that was accompanied by decreases in blood VLDL, NEFA, and glucose, liver weight and lipid content, and eWAT weight. The reversibility of this weight loss without apparent decreases of mouse viability suggests the potential value of liver GCN1 as a drug target for obesity or non-alcoholic fatty liver disease leading to insulin resistance, the most prevalent non-communicable disease in the world.

## 4. Materials and Methods

### 4.1. Animals

C57BL/6J mice were obtained from Clea Japan (Tokyo, Japan). CreERT2 knock-in mice [15] (Stock number 0008463) were obtained from Jackson laboratory (Bar Harbor, ME, USA). The mice were fed ad libitum CE-2 diet (Clea Japan) and maintained under temperature- and humidity-controlled rooms on a 12-h light-dark cycle (light from 8:00 to 20:00). The mice were sacrificed around 14:00 to 16:00, and the serum and tissue samples were collected. All the mouse experiments were approved by the Committee for the Ethics of Animal Experimentation of Hirosaki University and carried out according to the Principles of Laboratory Animal Care (National Institutes of Health, publication no. 85-23, revised 1985) and the Guideline for Animal Experimentation of Hirosaki University. The male mice without fasting were used for all the experiments except for the fertility analysis.

### 4.2. Generation of Gcn1 CKO Mice

*Gcn1* floxed mice were generated by the transfection of Cas9/single-guide RNA (sgRNA) expression vector pX330 targeting *Gcn1* exon 2 as described previously [2] and donor vector pFlox to introduce loxP. pX330 for Gcn1 exon 2 was constructed by insertion of oligo nucleotides for Gcn1 intron 1 (5′-TGC TGC TGA TGT GAG TGC GG-3′) or intron 2 (5′-GCC TGT AGG GAC TGT TCT CG-3′). To construct pFlox for Gcn1 exon 2, homologous regions of Gcn1 intron 1, exon 2, and intron 3 were amplified with following primers and inserted into pFlox plasmid; Left_arm_F (5′-CTG GGA TCC ACG TTT GTG TGC TAA CTC AGT TGA AGG AGG CTG T-3′), Left_arm_R (5′-TAC GAA GTT ATG TTT CAC TCA CAT CAG CAG CAC TAA CTG CAC C-3′), Central_arm_F (5′-CGA AGT TAT GGC GCG CCC GGG GGT GGG GCC GCC TG-3′), Central_arm_R (5′-TAT CGA ATT GGC GCG TCG GGG CGC ATT AGA AAA CAC ACA TCT C-3′), Right_arm_F (5′-CGA AGT TAT GCG GCC GAA CAG TCC CTA CAG GCG CTT GCA TTT T-3′), and Right_arm_R (5′-TTC GAA TTC GCG GCC AAC CTG CAT TTG TCA ACT GTG GGA AAT A-3′).

The pronuclei of fertilized C57BL/6J eggs were injected with pX330 and pFlox plasmids. F0 mice harboring both loxP sites and no plasmid integration were screened by PCR and restriction fragment length polymorphism (RFLP). Left or right loxP was amplified using the following primers and digested with *AscI* or *EcoRV*, respectively; Left_F (5′-CGG GCT GCA GGA ATT AAA AGA AGG CAT TGT TCA TTG TCT GTG C-3′), Left_R (5′-GCT TGA TAT CGA ATT CCC GGA GTG TTA GAG GAC TGA AAG CTA C-3′), Right_F (5′-CTG CTC GTC TGC TTC TGA CTA CAC CTT T-3′), and Right_R (5′-TAT CGA TGC AGT GTC AAG CAG AAC AAC T-3′). Insertion of both loxP sites were confirmed by sequencing. Genomic integration of donor vector and Cas9 vector was detected by PCR; Amp_F (5′-TTG CCG GGA AGC TAG AGT AA-3′), Amp_R (5′-TTT GCC TTC CTG TTT TTG CT-3′), Cas9_F (5′-AGT TCA TCA AGC CCA TCC TG-3), and Cas9_R (5′-GAA GTT TCT GTT GGC GAA GC-3′). Gcn1 exon 2 floxed (Gcn1f/+) mice were backcrossed with C57BL/6J or crossed with CreERT2 knock-in (*R26^Cre^*^/^*^+^*) mice at least 4 times.

Genotyping of the right loxP insertion was routinely detected by PCR using following primers; Gcn1_right_F (5′-GTG GCG TCC AGC TAA GTA CC-3′), and Gcn1_right_R (5′-AGG GAG GGA TGG AAG GTA GG-3′). Wild-type and floxed alleles are amplified as 279 and 341 bp bands. Left loxP insertion was detected by PCR using following primers; Gcn1_left_F (5′-GGG TAT GTA CGT GTG CGT GA-3′) and Gcn1_left_R (5′-CCC AAA AGT TCA CCC CTG GT-3′). The wild-type and floxed alleles were amplified as 269 and 315 bp bands. The deleted allele was detected by using Gcn1_left_F and Gcn2_right_R primers which was amplified as 395 bp bands, whereas the undeleted floxed allele was amplified as 1773 bp bands.

To induce CreERT2 activation, the male mice at 5–6 weeks old were intraperitoneally injected 10 times (5 consecutive days with 2-days interval) with 75 mg/kg tamoxifen that was dissolved in corn oil (3.75 mL/kg) (Appendix A). The mice were kept with monitoring of the food intake, water intake, and excretion, and sacrificed at day 14 (2 days after the last injection) or at day 42 (30 days after the last injection). For analysis of repeated tamoxifen injection to established CKO mice (Figure 4), the mice at day 43 were injected with tamoxifen or the vehicle in the same schedule and analyzed at day 56 and day 84.

### 4.3. Tamoxifen Metabolite Quantification

The serum concentration of tamoxifen and its metabolites, 4-OHTA and endoxifen were quantified by liquid chromatograph-mass spectrometry (LC-MC/MS) according to previous report [62] with minor modifications. The samples were prepared using solid-phase extraction kit ISOLUTE PLD+ (Biotage, Uppsala, Sweden). Briefly, 50 µL of serum and 200 µL of acetonitrile were mixed and incubated for 25 min, and the precipitated proteins were filtered using positive pressure manifold under 5 psi nitrogen gas. The soluble fraction was dried up by nitrogen gas and reconstituted in 50 μL acetonitrile–4 mM ammonium formate buffer pH 3.5 (3:7, *v*/*v*). LC-MS/MS was performed using an HPLC system ExionLC AD (AB Sciex, Tokyo, Japan)that was coupled to a QTRAP6500+ mass spectrometer (AB Sciex) with electrospray ionization (ESI), positive mode. A total of 15 µL microliters of the sample extract were injected onto an HPLC C18 column (Zorbax Eclipse XDB-C18 column, 3 × 100 mm, 3.5 µm, Agilent, Santa Clara, CA, USA) with a guard column (ZORBAX SB-C18, 3 × 100 mm, 1.85 µm, Agilent) at 55 °C and analyzed using acetonitrile gradient employing 5 mM ammonium formate pH 3.5 (solvent A) and acetonitrile (solvent B). The metabolites were identified and quantified using multiple reaction monitoring (MRM). Calibration curves were established using standards of tamoxifen (Sigma-Aldrich, St. Louis, MO, USA), 4-OHTA (Sigma-Aldrich) and endoxifen (E:Z = 1:1, Namiki shoji, Tokyo, Japan). Yields of the sample preparation were adjusted with the external standards that were diluted in wild-type mice serum.

### 4.4. Histological Analysis

The mice were anesthetized by i.p. injection of 0.2 mL/10 g body weight with a mixture of medetomidine (1 mg/mL), midazolam (5 mg/mL), and butorphanol (5 mg/mL). The tissues were fixed in the Mildform 10N (Wako Pure Chemicals) and embedded in paraffin and then were sectioned with 4 μm thick. Hematoxylin and eosin (H&E) staining was carried out as described previously [63]. Periodic acid-Schiff (PAS) staining was carried out as described previously [64].

For Oil Red O staining, the frozen liver samples that were embedded in O.C.T. compound (Sakura, Tokyo, Japan) were sectioned at 10 µm thick by a cryostat and then subjected to Oil Red O staining (Sigma-Aldrich) as described previously [65,66]. Briefly, the sections were immersed in 60% isopropanol, stained with Oil Red O solution (0.375% Oil Red O dissolved in 60% isopropanol, prepared and filtered each time freshly) for 5 min, and then washed in 60% isopropanol and distilled water, successively. The sections were covered with water-soluble medium (Dako, Carpinteria, CA, USA) and analyzed by a microscope BZ-X700 (Keyence, Osaka, Japan). The images were acquired at 10 regions per sample and Oil Red O-positive foci were selected and the area was measured using Image J software.

The frozen liver sections were subjected to COX/SDH staining as described previously [67,68,69]. The sections were incubated in freshly prepared COX or SDH solution at 37 °C for 30 min, followed by three times wash in 50 mM phosphate buffer (pH 7.4) for 5 min each. The COX solution contained 20 mM DAB (Takara Bio, Shiga, Japan), 2 mM cytochrome C from horse heart (Nacalai Tesque, Kyoto, Japan), and 20 µg/mL catalase from bovine liver (Fujifilm, Tokyo, Japan) in 50 mM phosphate buffer (pH 7.4). The SDH solution contained 0.5 mM nitro blue tetrazolium (Fujifilm), 42 mM disodium succinate (Fujifilm), 0.4 mM phenazine methosulfate (Tokyo Chemical Industry, Tokyo, Japan), and 2 mM sodium azide (Wako) in 50 mM phosphate buffer (pH 7.4). Unspecific signals of COX and SDH staining were evaluated by the addition of sodium azide and malonic acid (Fujifilm), respectively.

### 4.5. Mouse Embryonic Fibroblast Preparation

Mouse embryonic fibroblast (MEF) of *Gcn1^fl^*^/*fl*^::*R26^Cre^*^/*Cre*^ mouse were prepared as described previously [2]. The MEFs were maintained in Iscove’s modified Dulbecco’s medium (Sigma-Aldrich) that was supplemented with 10% fetal bovine serum (FBS, Life Technologies, Carlsbad, CA, USA), 100 U/mL Penicillin-Streptomycin (Thermo Fisher Scientific, Waltham, MA, USA), and 2 mM L-glutamine (Sigma-Aldrich), under 5% CO_2_, 100% humidity at 37 °C. The MEFs were treated with 1 µM 4-OHTA for 8 days to make Gcn1 CKO MEF; MEFs that were treated with the vehicle (0.1% DMSO) were prepared as a negative control (NC). The leucine-deficient medium was made by Cell Science & Technology Institute (Sendai, Japan) as Dulbecco’s modified Eagle medium (D’MEM, Sigma-Aldrich) composition and supplemented with 10% dialyzed FBS, 4.5 g/L glucose and 100 U/mL Penicillin-Streptomycin. The high glucose D’MEM that was supplemented with 10% FBS and 100 U/mL Penicillin-Streptomycin were used a control medium.

Cell proliferation was analyzed by using Cell Counting Kit-8 (CCK-8, Dojindo, Kumamoto, Japan) and absorption at 450 nm was used to calculate relative cell number.

### 4.6. Mouse Primary Hepatocyte Preparation

The primary hepatocytes were isolated from *Gcn1* CKO and NC mice at day 42 (11–12 week-old) as described previously [70]. The isolated hepatocytes were seeded at a density of 8 × 10^4^ cell/cm^2^ in dishes or plates that were coated with Cellmatrix type I-C (Kurabo, Osaka, Japan).

### 4.7. Reverse Transcriptase-Quantitative PCR (RT-qPCR)

According to the manufacturer’s instruction, the total RNA was extracted from tissue samples that were stored frozen using TRIzol RNA Isolation Reagents (Thermo Fisher Scientific), cDNA was synthesized from the total RNA using PrimeScript IV 1st strand cDNA Synthesis Mix (Takara Bio). Real-time PCR was performed by a CFX96 thermal cycler (Bio-Rad, Richmond, CA, USA) using TB Green Premix Ex Taq II (Takara Bio) and primers described; ASNS-F (5′-GGC CCT TGT TTA AAG CCA TGA-3′) and ASNS-R (5′-AAG GGA GTG GTG GAG TGT TTT-3′), Mthfd2-F (5′-CCT ACA GCC CTT CCA CCT G-3′) and Mthfd2-R (5′-GAG GCC ACC CAC TCT TCC-3′), xCT-F (5′-TGG GTG GAA CTG CTC GTA AT-3′) and xCT-R (5′-AGG ATG TAG CGT CCA AAT GC-3′), Cpt1a-F (5′-GAC TCC GCT CGC TCA TTC C-3′) and Cpt1a-R (5′-GGC AGA TCT GTT TGA GGG CT-3′), eEF1b-F (5′-ATT ACC TGG CGG ACA AGA GC-3′) and eEF1b-R (5′-AAC GTA GGG CAT GAC ACA GG-3′), Hadhb-F (5′-GGC GGA CGT TTG TCA GTC T-3′) and Hadhb-R (5′-TGA AAT CTG CCT GTG GGG AAA-3′), Mccc1-F (5′-GGG ACT GAC TTG GTG GAG TG-3′) and Mccc1-R (5′-CAG CCC CTG GCA TGA AGT TA-3′), Mccc2-F (5′-TAT TTG GGT ACC CCG TTG GC-3′) and Mccc2-R (5′-CAC CAT CCT TGG CAA TCC CT-3′), Cypa-F (5′-AAG ACT GAA TGG CTG GAT GG-3′) and Cypa-R (5′-AGC TGT CCA CAG TCG GAA AT-3′), UCP1-F (5′-CAC GGG GAC CTA CAA TGC TT-3′) and UCP1-R (5′-ACA GTA AAT GGC AGG GGA CG-3′), PPARa-F (5′-TGC CTT CCC TGT GAA CTG AC-3′) and PPARa-R (5′-TGG GGA GAG AGG ACA GAT GG-3′), Acc1-F (5′-CCC ATA GCC TGT GGA ACC CC-3′) and Acc1-R (5′-TCC CTT TGC CCT GGT CAA GC-3′), HSL-F (5′-GGA GCT CCA GTC GGA AGA GG-3′) and HSL-R (5′-GTC TTC TGC GAG TGT CAC CA-3′), LPL-F (5′-CCA GCT GGG CCT AAC TTT GA-3′) and LPL-R (5′-AAC TCA GGC AGA GCC CTT TC-3′). *Adiponectin* was detected by using primers that were reported previously [71]. *PPARγ* and *SREBP1* were detected using primers as described [58]. The data were analyzed by the ΔΔCt method and were normalized to the expression of the internal control *Cypa* gene.

### 4.8. Western Blot Analysis

The mice tissues were lysed in Laemmli’s sample buffer containing cOmplete, EDTA-free protease inhibitor cocktail (Roche, Mannheim, Germany) and were homogenized using Minilys (Bertin Technologies, Montigny-le-Bretonneux, France) at 5000 rpm for 60–90 s, and the protein concentration was measured by BCA protein assay kit (Thermo Fisher Scientific) according to the manufacturer’s instruction. The protein samples were supplemented with 5% 2-mercaptoethanol, denatured by boiling, and separated on SDS-PAGE gel and transferred to a polyvinylidene difluoride (PVDF) membrane (Millipore, Billerica, MA, USA). The membrane was blocked using 3% skim milk and then was incubated with Gcn1 (1:10000; ab86139, Abcam), D-glyceraldehyde-3-phosphate dehydrogenase antibody (GAPDH, 1:5000; GTX100118, GeneTex, Irvine, CA, USA) overnight at 4 °C followed by 1 h incubation with HRP-conjugated goat anti-rabbit IgG (H+L) secondary antibody (1:10000; G21234, Invitrogen) at room temperature. Immobilon Forte Western HRP Substrate (Millipore) was used for chemiluminescence detection. The immunoreactive band intensities were measured by using Image J software and normalized by GAPDH.

### 4.9. Biochemical Marker Measurements

Whole blood was collected from the anesthetized mice via cardiac puncture and centrifuged at 1200× *g* for 10 min and the serum was aliquoted and stored at −25 °C. The serum was subjected to measurements of aspartate aminotransferase (AST), alanine aminotransferase (ALT), lactate dehydrogenase (LDH), creatine phosphokinase (CPK), total bilirubin (T-bil), blood urea nitrogen (BUN), and glucose (GLU) using Spotchem II Stat-1 and Spotchem II Glucose reagents (Arkray, Shiga, Japan) and an automatic biochemical analyzer Spotchem EZ SP-4430 (Arkray). The blood insulin was quantified using Mouse/Rat Insulin ELISA Kit (Morinaga, Tokyo, Japan) according to manufacturer’s instruction. Ketone body was determined using EnzyChrom Ketone Body Assay Kit (BioAssay Systems, CA, USA). Serum glucagon and non-esterified fatty acid (NEFA) were quantified using Glucagon ELISA kit (Mercodia, Uppsala, Sweden) and LabAssay NEFA kit (Wako, 279-75401), respectively, according to the manufacturer’s instruction. Lipoprotein profile of serum was performed by LipoSEARCH service (Immuno-Biological Laboratories, Gunma, Japan) to measure the cholesterol and triglyceride in chylomicron, VLDL, LDL, and HDL.

### 4.10. Liver Glycogen Measurement

The liver tissues were homogenized in ultrapure water (25 μL/mg liver) on ice, boiled for 5 min, and centrifuged at 13,000× *g* for 5 min. The supernatant was diluted and subjected to the Glycogen assay kit (MAK016, Sigma-Aldrich) according to the manufacturer’s instruction. Absorbance at 570 nm was quantified using a microplate reader (Bio-rad) then the glycogen concentration was calculated using a standard curve.

### 4.11. Fecal Glucose Measurement

The fecal samples were collected at 7:30−8:00 and stored at −80 °C. The fecal glucose was prepared as described by Shirai et al. [72] and quantified by LabAssay Glucose kit (Wako, 638-50971) according to the manufacturer’s instruction.

### 4.12. ATP Measurement

The liver tissues were homogenized in ultrapure water (100 μL/mg liver) on ice and centrifuged at 1000× *g* for 10 min. The supernatant was diluted 7-fold and extracted ATP according to the manufacturer’s instruction. The luminescence was measured using GloMax 20/20 (Promega, Madison, WI, USA).

### 4.13. Oxygen Consumption Rate (OCR) Measurement

Primary mouse hepatocytes were suspended at the density of 2000 cell in 80 µL culture medium and inoculated onto miniplate which had been coated with Cellmatrix type I-C (Kurabo). The culture media were replaced to Seahorse XF DMEM medium (Agilent) supplemented with 10 mM glucose, 1 mM pyruvate, and 2 mM L-glutamine prior to the assay. The OCR assay was performed using Seahorse XFp analyzer and Seahorse XF Cell Mito stress kit (Agilent) according to manufacturer’s instruction. The following reagents were sequentially dispensed; 1.5 µM oligomycin, 1 µM FCCP, and 2 µM rotenone/antimycin A.

### 4.14. Proteome Analysis

Proteomic analyses of the liver was performed using a nanoLC Eksigent 400 system coupled to an TripleTOF6600 mass spectrometer (AB Sciex, Framingham, MA, USA). Briefly, the proteins (20 μg) that were extracted from the liver in Laemmli buffer were precipitated with acetone. The precipitates were denatured with 50% trifluoroethanol and reduced using 4 mM dithiothreitol. The free cysteine residues were alkylated before trypsinization. The tryptic peptides were analyzed using the mass spectrometer that was operated in information-dependent acquisition mode. The acquired spectra were searched against the UniProt database using a Paragon algorithm that was embedded in ProteinPilot 5.0.1 software program (AB Sciex). Positive identifications were considered when the identified proteins and peptides reached a 1% local false discovery rate. Each sample was also assayed in data-independent acquisition mode, and the proteins were quantified using the information-dependent acquisition data as a library.

### 4.15. Statistical Analysis

Statistical analysis was performed using GraphPad Prism 7.0 software. Two tailed Student’s *t*-tests and one-way ANOVA were used for analysis of differences between two and three groups, respectively. Multiple group comparison was analyzed by two-way ANOVA and the Tukey or Sidak method as described in the figure legends. The data are presented as the means with standard error of the mean (SEM) values. Statistical significance was considered as *p* < 0.05.

For the proteomics data, we used the Welch’s *t*-test in MarkerView 1.3 software (AB Sciex) to determine statistically significant changes of proteins between the experimental groups. *p* < 0.05 were considered to indicate statistical significance. A principal component analysis (PCA) and OPLS-DA were performed using Simca software program (Infocom Corp., Tokyo, Japan). Pareto scaling was applied to the normalized peak area values acquired by SWATH prior to the analyses, which represents a compromise between the extremes of no scaling and unit variance scaling and involves dividing each spectral variable by the square root of its standard deviation after first centering. Pathway analysis of differentially regulated proteins were performed using IPA (Qiagen). A *p*-value reflecting the probability that the association was explained by chance alone was calculated by Fisher’s exact test. The proteins altered in CKO liver with p_1_(corr) > 0.6 or < −0.6 by OPLS-DA were analyzed by IPA.

## Figures and Tables

**Figure 1 ijms-23-03201-f001:**
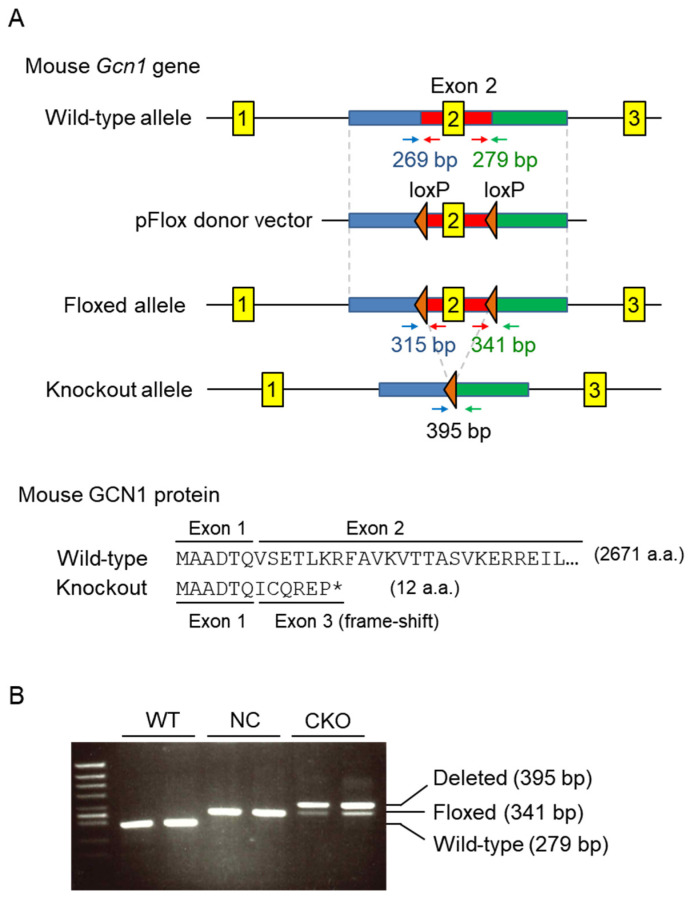
Tamoxifen-induced *Gcn1* exon2 conditional knockout. (**A**) A schematic diagram explaining the method for establishing *Gcn1* CKO mice. The LoxP sites were inserted by CRISPR/Cas9 and pFlox donor vector targeting *Gcn1* exon 2. The genotyping primers surrounding loxP sites are indicated by arrows. Deletion of *Gcn1* exon 2 results in a frame-shift with premature stop codon (*) in exon 3. (**B**) Genotyping of the wild-type (WT) mice, *Gcn1^fl^*^/*fl*^::*R26*^+/+^ mice that were treated with tamoxifen (NC), and *Gcn1^fl^*^/*fl*^::*R26^Cre^*^/*Cre*^ mice that were treated with tamoxifen (CKO). Genomic DNA samples that were extracted from the tail were used to distinguish right loxP in the WT allele (279 bp), floxed (341 bp), and deleted allele in CKO (395 bp).

**Figure 2 ijms-23-03201-f002:**
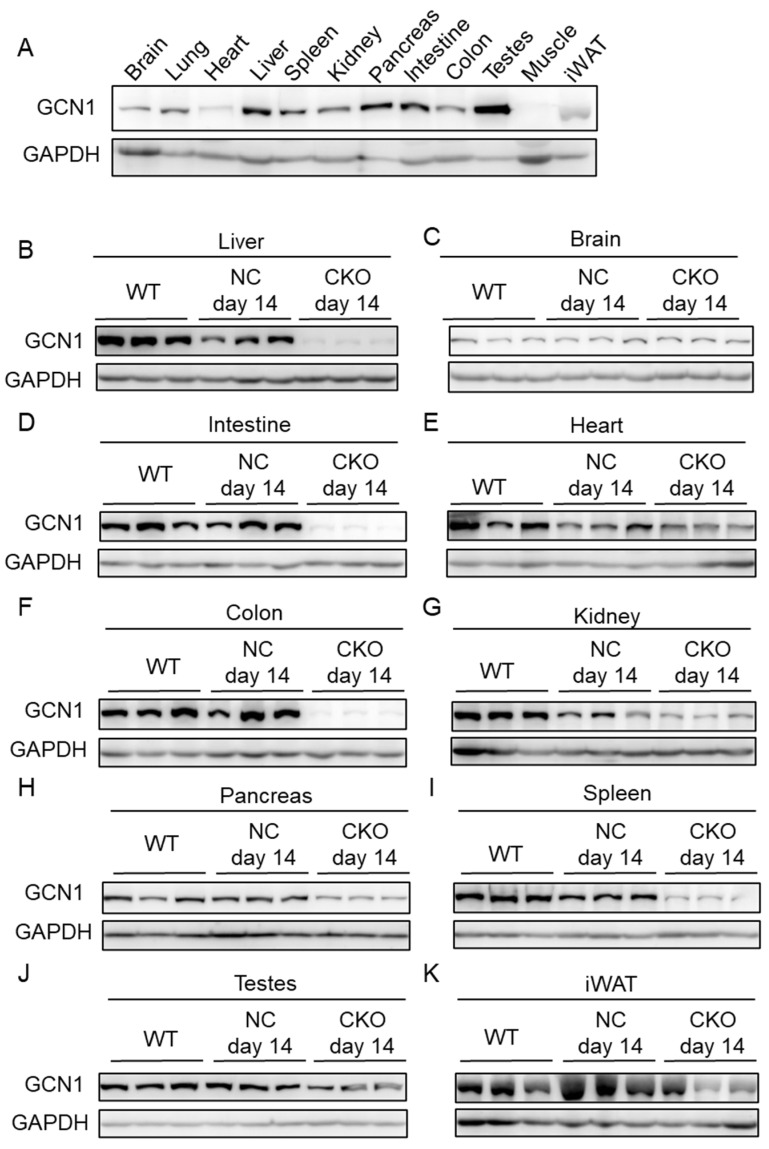
Expression of GCN1 in the tissues of wild-type and CKO mice. (**A**) GCN1 protein expression in the tissues of wild-type male mice (six weeks old). GCN1 and the loading control, GAPDH, were detected by immunoblotting. (**B**–**K**) GCN1 expression in untreated wild-type mice (day 0), and tamoxifen-administrated *Gcn1^fl^*^/*fl*^::*R26*^+/+^ (NC, day 14) and *Gcn1^fl^*^/*fl*^::*R26^Cre^*^/*Cre*^ (CKO, day 14) mice. GCN1 expression was examined in the liver (**B**), brain (**C**), intestine (**D**), heart (**E**), colon (**F**), kidney (**G**), pancreas (**H**), spleen (**I**), testes (**J**), and inguinal white adipose tissue (iWAT; (**K**)).

**Figure 3 ijms-23-03201-f003:**
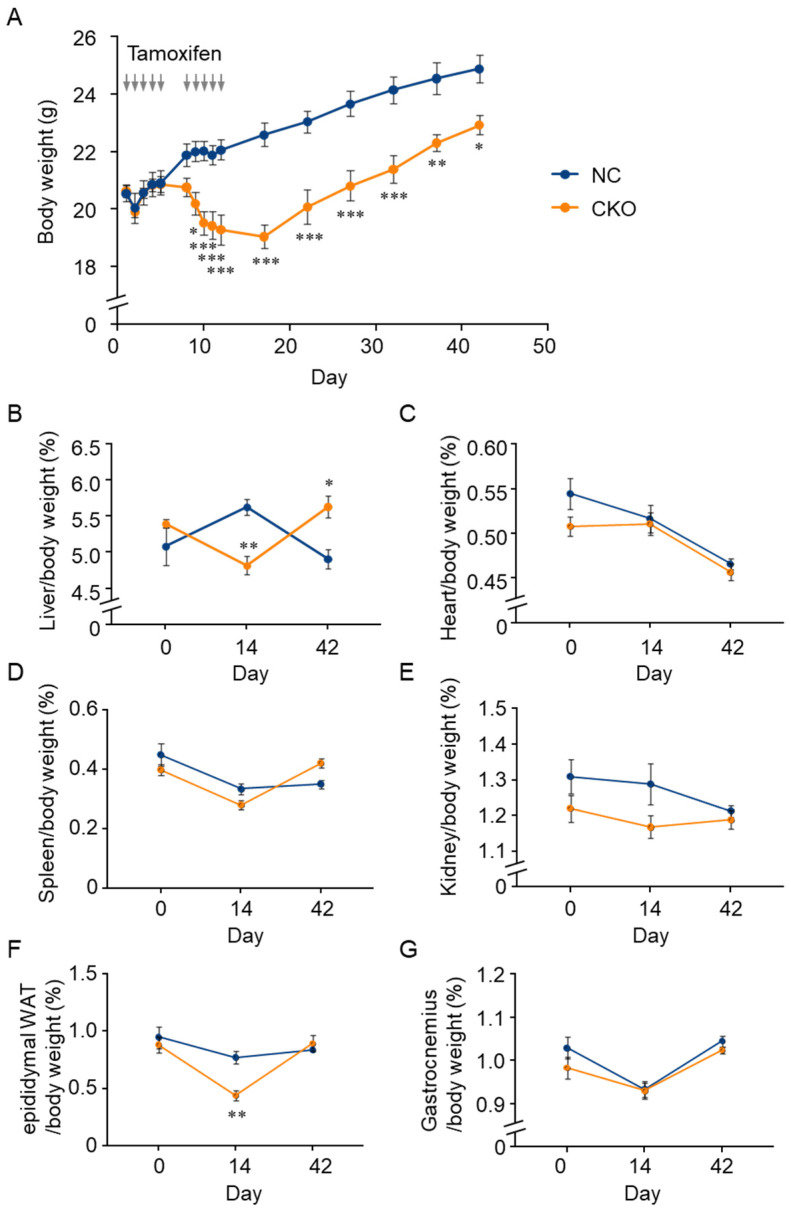
Body weight change of tamoxifen-induced *Gcn1* CKO mice. (**A**) Body weight change of *Gcn1^fl^*^/*fl*^ and *Gcn1^fl^*^/*fl*^::*R26^Cre^*^/*Cre*^ mice that were treated with tamoxifen. Male mice at five to six weeks old were injected 10 times with 75 mg/kg tamoxifen as indicated by the gray arrows, which generated negative control (NC) and conditional knockout (CKO), respectively. The body weight was recorded every day during the injection period, and then every five days until day 42 (*n* = 6 per group). The data are presented as the mean ± SEM. Statistical significance was analyzed by two-way ANOVA and Sidak method. (**B**–**G**) The tissue weight change in CKO mice. The weight of the liver (**B**), heart (**C**), spleen (**D**), kidney (**E**), epididymal WAT (**F**), and gastrocnemius (**G**) was normalized to body weight (*n* = 6, except for gastrocnemius in the NC group, *n* = 5). The data are presented as the mean ± SEM. Statistical significance was analyzed by two-way ANOVA and Tukey’s test. * *p* < 0.05, ** *p* < 0.01, *** *p* < 0.001.

**Figure 4 ijms-23-03201-f004:**
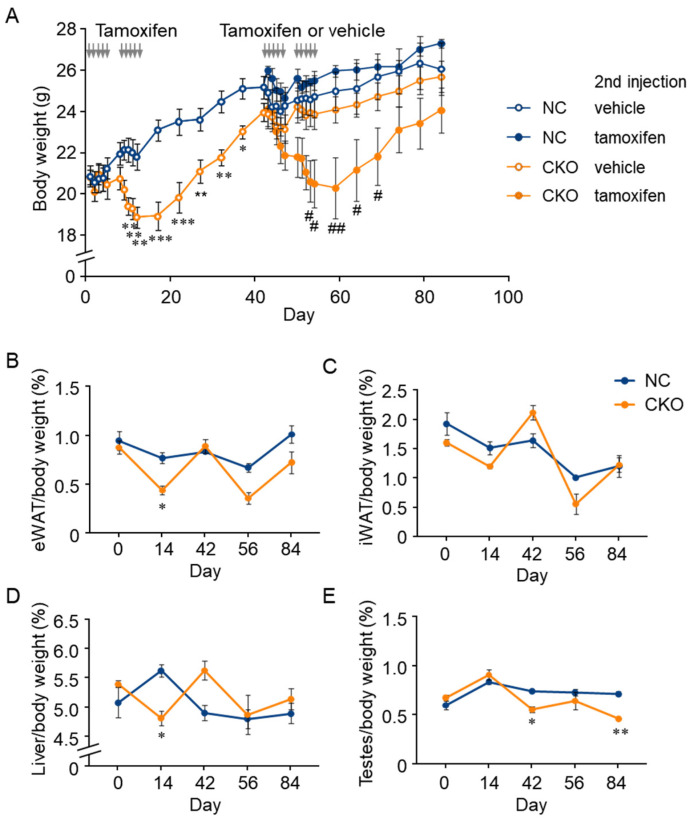
Combination of GCN1-deletion and tamoxifen-treatment induces the loss of body weight gain. (**A**) Body weight change of CKO mice that were treated with two rounds of tamoxifen injections. The *Gcn1* CKO (*n* = 6) or NC (*n* = 6) mice were randomly separated (*n* = 3 per group), and then subjected to a second round of injections of tamoxifen or control vehicle at day 43 by the same regimen. Statistical analysis was carried out by two-way ANOVA and Tukey’s test. * *p* < 0.05, ** *p* < 0.01, *** *p* < 0.001 compared with NC/tamoxifen; ^#^ *p* < 0.05, ^##^ *p* < 0.01 compared with the CKO/vehicle. (**B**–**E**) Tissue weight change upon two rounds of injections of tamoxifen or control vehicle were analyzed in the epididymal WAT (**B**), inguinal WAT (**C**), liver (**D**), and testes (**E**). The data in (**B**–**E**) are presented as the mean ± SEM (*n* = 3) and statistically analyzed by two-way ANOVA and Tukey’s test. * *p* < 0.05, ** *p* < 0.01 compared with NC group.

**Figure 5 ijms-23-03201-f005:**
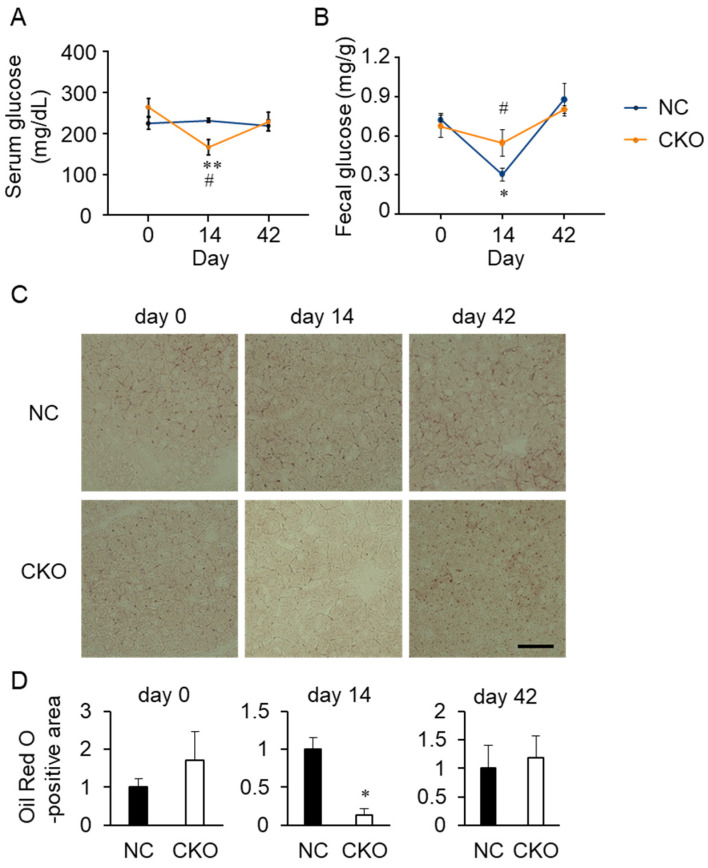
Impaired glucose and fat metabolism in tamoxifen-induced *Gcn1* CKO mice. (**A**) The serum glucose in the NC and CKO mice were measured as described in the Materials and Methods. The data are presented as the mean ± SEM of NC/day 0 (*n* = 3), CKO/day 0 (*n* = 3), NC/day 14 (*n* = 7), CKO/day 14 (*n* = 7), NC/day 42 (*n* = 4), and CKO/day 42 (*n* = 3). Statistical analysis was carried out by two-way ANOVA and Tukey’s test. ** *p* < 0.01 compared with day 0, ^#^ *p* < 0.05 compared with NC. (**B**) Fecal glucose was quantified as described in the Materials and Methods (*n* = 3). Statistical analysis was carried out by two-way ANOVA and Tukey’s test. * *p* < 0.05, compared with day 0; ^#^ *p* < 0.05 compared with NC. (**C**) Frozen liver sections were subjected to Oil Red O staining to examine the lipid accumulation. Scale bar, 50 µm. (**D**) The area of Oil Red O-positive region per image was quantified by ImageJ and normalized to NC. The data are expressed as the mean ± SEM (*n* = 3 per group). Statistical analysis was carried out by *t*-test. * *p* < 0.05.

**Figure 6 ijms-23-03201-f006:**
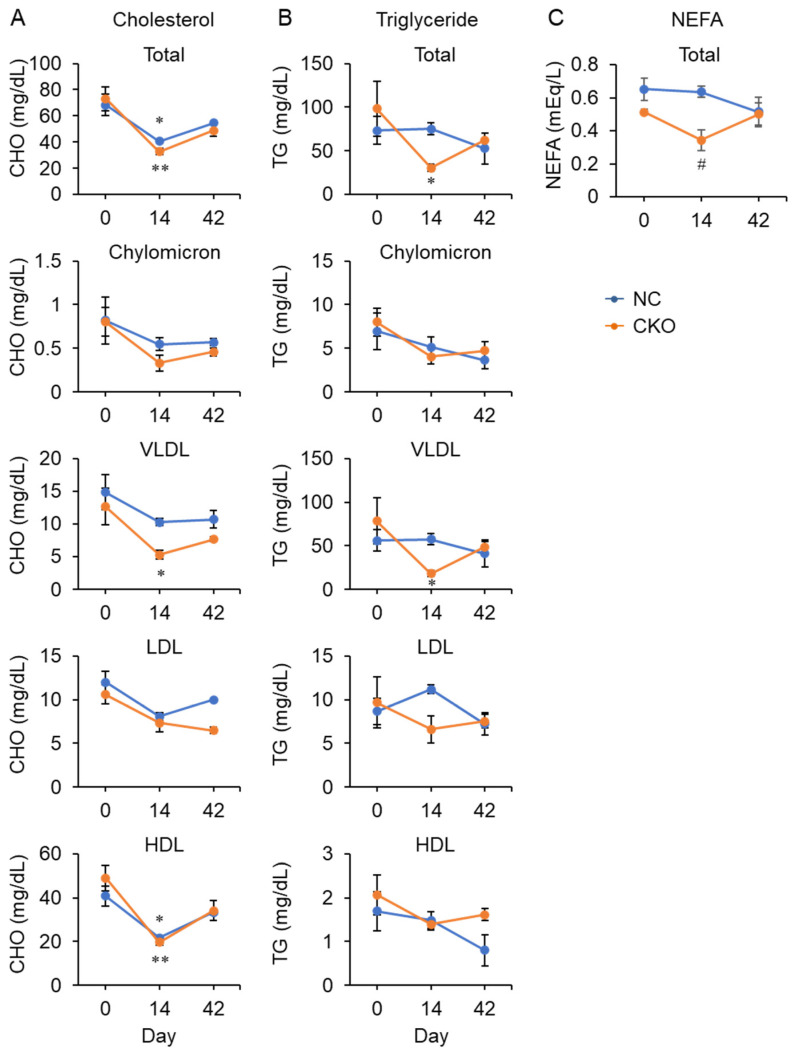
Alteration of the lipoprotein in the *Gcn1* CKO mouse. (**A**–**C**) Amounts of cholesterol (**A**), triglyceride (**B**), and NEFA (**C**) were quantified in the total serum and in fractions of chylomicron, VLDL, LDL, and HDL. The data are expressed as the mean ± SEM of NC/day 0 (*n* = 3), CKO/day 0 (*n* = 3), NC/day 14 (*n* = 4), CKO/day 14 (*n* = 4), NC/day 42 (*n* = 3), and CKO/day 42 (*n* = 2) for cholesterol and triglyceride (**A**,**B**) and n = 4 per group for NEFA (**C**). Statistical analysis was carried out by two-way ANOVA and Tukey’s test. * *p* < 0.05, ** *p* < 0.01 compared with day 0, ^#^ *p* < 0.05 compared with NC.

**Figure 7 ijms-23-03201-f007:**
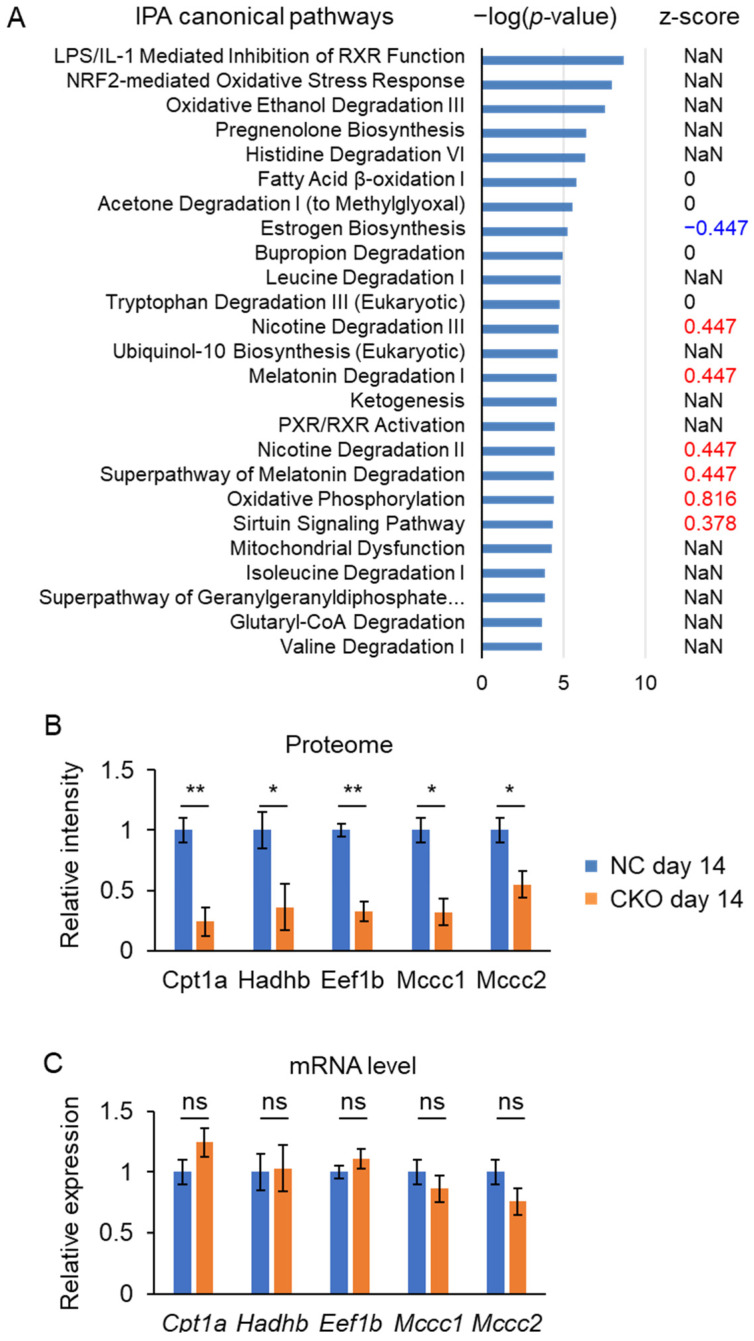
Proteomic analysis of *Gcn1* CKO mouse liver. (**A**) Top 25 canonical pathways that were analyzed by IPA of 137 proteins correlated with a discriminant model in OPLS-DA correlation [p_1_(corr) > 0.6 or < −0.6]. −Log(*p*-value) and z-score (activation in red, inactivation in blue, and neutral and not a number, NaN in black). (**B**) Results of the proteomic quantification of proteins markedly decreased in CKO liver are expressed as the mean ± SEM (*n* = 4) and normalized by the value in the NC liver. (**C**) mRNA levels of transcripts that are listed in (**B**) were quantified by RT-qPCR and normalized by *Cypa*, with that in NC liver expressed as 1. The data are expressed as the mean ± SEM (*n* = 4 per group). The data were statistically compared by *t*-test. * *p* < 0.05, ** *p* < 0.01, not significant (n.s.). (**D**) Summarized pathways of lipid metabolism in the CKO mouse liver. Proteins significantly increased and decreased (except for Acads, *p* = 0.057) in CKO are shown in red and blue, respectively. Decreases in lipids that were determined by the other analyses are indicated by downward arrows. The pathways predicted to be attenuated are indicated by a gray arrow with a dashed line and increased are indicated by a bold arrow, respectively.

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
