# Peer review of "Inducible Systemic Gcn1 Deletion in Mice Leads to Transient Body Weight Loss upon Tamoxifen Treatment Associated with Decrease of Fat and Liver Glycogen Storage"

_ijms, 2022, doi:10.3390/ijms23063201_

Round 1

Reviewer 1 Report

Liu J et al create a Gcn1 floxed mice and using the Rosa 26 locus deleted the gene in the adult mice. This provides insight into the overall role of Gcn1.

Major concern I have with this study is that the author are downplaying the role in Gcn1 in the phenotype observed. Instead, they are concluding that tamoxifen is having the effect. However, they do not observe these changes in the Cre negative mice given tamoxifen, so it is clear that the observed effects are due to Gcn1 deletion. 

To conclusively state that the metabolic effects of Gcn1 deletion are due to its role in liver, the authors should consider creating a liver specific deletion of Gcn1 using Alb-Cre. Since this might be beyond the scope of this study, authors should isolate hepatocytes from the floxed mice and use in vitro Cre delivery to delineate the role of Gcn1 in liver hepatocytes that play a major role in metabolism. The possibility of utilizing cell specific cre should be discussed.

The authors should clearly state the gender of the mice utilized in the study to rule of any gender-based differences in the phenotype observed.

Author Response

Liu J et al create a Gcn1 floxed mice and using the Rosa 26 locus deleted the gene in the adult mice. This provides insight into the overall role of Gcn1.

Major concern I have with this study is that the author are downplaying the role in Gcn1 in the phenotype observed. Instead, they are concluding that tamoxifen is having the effect. However, they do not observe these changes in the Cre negative mice given tamoxifen, so it is clear that the observed effects are due to Gcn1 deletion.

Thank you for the comment. Although we also think the GCN1 deficiency is responsible for the observed phenotype, we concluded the phenotype is transiently and reversibly caused by a combination of Gcn1 CKO and tamoxifen treatment, but not by Gcn1 CKO alone. Since our manuscript have somewhat complicated parts, we modified the abstract and discussion and added brief summary in later part of the introduction.

To conclusively state that the metabolic effects of Gcn1 deletion are due to its role in liver, the authors should consider creating a liver specific deletion of Gcn1 using Alb-Cre. Since this might be beyond the scope of this study, authors should isolate hepatocytes from the floxed mice and use in vitro Cre delivery to delineate the role of Gcn1 in liver hepatocytes that play a major role in metabolism. The possibility of utilizing cell specific cre should be discussed.

We agree with the comment that analysis of liver-specific Gcn1 CKO mice and of GCN1 KO primary hepatocytes are useful to show the role of hepatic GCN1 in liver metabolism and exclude the extra-hepatic role of GCN1. As these experiments are beyond the scope of the current study, we added the possible future plans in the discussion that employ both liver-specific Gcn1 CKO mice and in vitro primary culture of hepatocytes.

The authors should clearly state the gender of the mice utilized in the study to rule of any gender-based differences in the phenotype observed.

Thank you for the suggestion. We added the gender in the results, the materials and methods sections, and in the figure legends.

Reviewer 2 Report

The current study investigates the association of the deletion of GNC1 protein and body weight loss in adult mice. It is impressive to see so much data being presented. However, to improve readability, it would be helpful to summarize the findings in layman's terms. Also, the authors could include extra details regarding the mouse population: the number of mice used in the study and the mice's age when the study started. The sequence of interventions could be described and included in the materials and methods. Also, the authors could comment why tamoxifen and not another compound was used to induce GNC1 knockout. 

Author Response

The current study investigates the association of the deletion of GNC1 protein and body weight loss in adult mice. It is impressive to see so much data being presented. However, to improve readability, it would be helpful to summarize the findings in layman's terms. Also, the authors could include extra details regarding the mouse population: the number of mice used in the study and the mice's age when the study started.

Thank you for the comment. We modified the abstract to be read more easily and added brief findings at the last part of the introduction. Although the number of mice used in the experiments was indicated in figure legends, some ambiguous expressions (such as n = 5–6) were corrected. The age of mice was also added in the figure legends.

The sequence of interventions could be described and included in the materials and methods. Also, the authors could comment why tamoxifen and not another compound was used to induce GNC1 knockout.

Thank you for the suggestion. We added the detailed explanation of mouse interventions in the materials and methods section, “4.2. Generation of Gcn1 CKO mice”.

We utilized tamoxifen-dependent CreERT2 inducible knockout system for the generation of inducible GCN1 KO as other systems such as interferon-inducible system cause more widespread stress response in the mice compared to the tamoxifen system. Instead of tamoxifen, we also used raloxifene to induce Gcn1 CKO as pilot experiment (n = 1) and observed that the injections of 150 or 300 mg/kg raloxifene decreased the body weight as well. However, we performed no further experiment because of the higher dose required to induce gene deletion and low solubility of raloxifene as well as limited information compared to tamoxifen.

Round 2

Reviewer 1 Report

The authors answered all the questions to my satisfaction. I have no new comments